# Novel Development of Nanoparticles—A Promising Direction for Precise Tumor Management

**DOI:** 10.3390/pharmaceutics15010024

**Published:** 2022-12-21

**Authors:** Dengke Zhang, Qingqing Tang, Juan Chen, Yanghui Wei, Jiawei Chen

**Affiliations:** 1Department of Surgery, The Eighth Affiliated Hospital, Sun Yat-sen University, Guangzhou 510275, China; 2Department of Medicine & Rehabilitation, Tung Wah Eastern Hospital, Hong Kong, China

**Keywords:** nanoparticles, tumor-targeting, antitumor drug

## Abstract

Although the clinical application of nanoparticles is still limited by biological barriers and distribution, with the deepening of our understanding of nanoparticles over the past decades, people are gradually breaking through the previous limitations in the diagnosis and treatment of tumors, providing novel strategies for clinical decision makers. The transition of nanoparticles from passive targeting to active tumor-targeting by abundant surface-modified nanoparticles is also a development process of precision cancer treatment. Different particles can be used as targeted delivery tools of antitumor drugs. The mechanism of gold nanoparticles inducing apoptosis and cycle arrest of tumor cells has been discovered. Moreover, the unique photothermal effect of gold nanoparticles may be widely used in tumor therapy in the future, with less side effects on surrounding tissues. Lipid-based nanoparticles are expected to overcome the blood–brain barrier due to their special characteristics, while polymer-based nanoparticles show better biocompatibility and lower toxicity. In this paper, we discuss the development of nanoparticles in tumor therapy and the challenges that need to be addressed.

## 1. Introduction

Among cancer treatment strategies, surgery, radiotherapy, chemotherapy, and immunotherapy are the most basic methods. With the continuous standardization and promotion of cancer treatment plans, the overall mortality of cancer has a downward trend; the 29% overall mortality rate of cancer decreased compared with the peak in 1991, but the total number of deaths is still large, with 609,360 cancer deaths being estimated to occur in America in 2022 [1]. The large number of cancer-associated mortalities is partly due to the imprecision of therapeutic methods and the caused side effects. For example, during radiotherapy, radiation can induce pneumonitis and dermatitis from damaging the healthy surrounding tissues; during systemic chemotherapy, normal tissues such as mucosal epithelium and bone-marrow hematopoietic tissues will be greatly affected because chemotherapy drugs cannot distinguish malignant cells from those cells [2]. This will lead to a decline in the therapeutic effect. Regarding this, more studies on tumor-targeting nanoparticles are warranted to provide new strategies for cancer treatment.

## 2. Targeting Mechanism of Nanoparticles

### 2.1. Passive Targeting of Nanoparticles

Molecules at the nanometer scale endow nanoparticles with a special property, which is the enhanced permeability and retention (EPR) effect [3]. For solid tumors, the capillary plexus in them is abnormal and disorganized; the density of these capillaries is increased, with significantly increased permeability, which allows nanoparticles with a molecular diameter of 100–800 nm to enter tumor tissue [4]. In contrast, the nutrient vascular endothelium of normal tissue is closely arranged. Meanwhile, the clearing function of lymphatic vessels in tumor tissue is impaired, causing a long-term retention of nanoparticles, which is one of the reasons for tumor immune escape [5]. As shown in Figure 1 below, the difference in the arrangement of vascular endothelial cells between tumor tissue and normal tissue results in this special property of tumor. However, this targeting mechanism is limited by the biological distribution in vivo, the nonspecific uptake of normal tissues, the degree of tumor vascularization, and the blood vessel flow. In the plasma environment of circulation, nanoparticles are adsorbed and opsonized by albumin, complement components, immunoglobulins, and other plasma proteins and subsequently internalized by the mononuclear phagocytic system (MPS), mainly in the liver, lymph nodes, and spleen [6]. It is suggested that nanoparticles that are 100–200 nm in size have an excellent EPR effect, which can also help them avoid the filter trap of liver and spleen to the greatest extent [5]. Moreover, for pancreatic ductal adenocarcinoma characterized by insufficient vascularization and nonsolid tumors, it is hard to target nanoparticles toward tumor cells [7].

### 2.2. Active Tumor-Targeting Mechanism of Nanoparticles

Many receptors, such as epidermal growth factor receptors (EGFRs), vascular endothelial growth factor receptors (VEGFRs), and folic acid receptors, which belong to normal cells were thriving, expressed in the surface of tumor cells; in addition, specific landmark polypeptides and proteins are also expressed on the surface. Therefore, these sites become targets for modified nanoparticles by corresponding ligands to “track” tumor cells. The following categories are the main identified targets.

#### 2.2.1. Human Epidermal Growth Factor Receptors (HERs)

HERs are tyrosine-kinase-coupled receptors which belong to the erythroblastic leukemia oncogene B family, meditating normal cell proliferation, tumor progression, and invasion [8]. HER1, also known as epidermal growth factor receptors (EGFRs), is overexpressed especially in non-small cell lung cancer [9]. HER2 is another member of this family. HER2 overexpression is estimated to occur in 15% to 30% of breast cancers and in about 10% to 30% gastric cancers. HER2 expression suggests a bad clinical outcome, which also makes it an attractive target for treatment and nanoparticles’ engineering design [10].

#### 2.2.2. Vascular Endothelial Growth Factor Receptors (VEGFRs)

The growth process of most solid tumors is accompanied by abnormal neovascularization, with vascular endothelial growth factors (VEGFs) induced by the hypoxic microenvironment in tumor [11]. In addition to the vasogenic role of VEGF/VEGFR, there is evidence that VEGFR 1 is involved in the colorectal carcinoma cells’ metastasis [11]. Moreover, VEGF/VEGFR inhibits the function of T lymphocytes and participates in the immune escape of tumors [12], making it a target of anticancer treatment.

#### 2.2.3. Fibroblast Growth Factor Receptors (FGFRs)

Fibroblast growth factor and its receptors regulate various functions, such as migration, proliferation, and differentiation of wide range cell types in healthy tissue [13]. In the major steps of tumor progression, such as angiogenesis, invasion, and metastasis of germ-cell neoplasms [14], breast cancer [15], and bladder cancer [16], the FGF/FGFR signal is excessive, suggesting that it is a potential anticancer target. In the US, two FGFR inhibitors, Erdafitinib [17] and Pemigatinib [18], have been approved in recent years for the treatment of metastatic urothelial carcinoma and unresectable cholangiocarcinoma, respectively.

#### 2.2.4. Nutrition-Related Receptors

The rapid proliferation of tumor cells often requires excessive nutrients, including albumin, folic acid, and iron. Albumin internalization relies on the binding with high affinity to the gp60 receptor on the surface of tumor endothelial cells, and the overexpression of secreted protein acidic rich in cysteine (SPARC) in different type of tumors also attracts albumin, promoting albumin accumulation inside the tumor cells [19]. Folic acid receptors are expressed in aggressive cancers, including ovarian cancer, breast cancer, and uterine adenocarcinoma, more than in normal tissue cells [20,21,22,23]. To meet the iron requirements of tumor cells, transferrin receptors are often expressed up to 10 times as much as normal tissues [24]. Albumin and transferrin can be used as ligands for drug targeting delivery with good biocompatibility. In addition, transferrin is a potential ligand which enables nanoparticles to cross the blood–brain barrier (BBB) and target glioma cells [25]. In pancreatic ductal adenocarcinoma (PDAC) progression, PDAC-derived cancer-associated fibroblasts (CAFs), which mediate peritumoral fibrosis and the creation of the drug physical barrier, are highly dependent on SLC7A11 for cystine uptake and glutathione synthesis; related experiments showed that the nanogene-silencing drug treatment against SLC7A11 inhibited the growth and metastasis of PDAC, CAF activation, and fibrosis [26].

#### 2.2.5. Tumor Specific Antigen, Protein and Peptide Receptor

Specific amino acid sequence composition on the tumor surface, including several receptors described above, can be recognized by B lymphocytes, and they then stimulate B cells to produce monoclonal antibodies with high affinity to tumors. In clinical settings, Perjeta and Herceptin targeting HER2 have been widely used in HER2 overexpressed breast cancer and gastric adenocarcinoma and have achieved excellent clinical results [27]. Nanoparticles binding with monoclonal antibodies can efficiently target tumors.

In the next part of this review, we focus on different kinds of nanoparticles and the potential of novel application scenarios they have shown in recent years.

## 3. Inorganic Nanoparticles

Many studies reported the role and potential of inorganic nanoparticles in oncology; this class of NP usually has unique physical characteristics, such as unique optical, electrical, thermal, and magnetic properties, which lead to their extensive applications in tumor diagnosis and treatment. Among them, Au nanoparticles and carbon nanoparticles are the most extensively studied nanoparticles.

### 3.1. Au Nanoparticles

As one of the most stable and least toxic metal NP formulations, Au NPs are widely used as a cancer-targeted drug-delivery system. D-P-HGNPs/21 is a sequential drug-delivery system which enters the tumor cells through the endocytic pathway and first releases MiR-21i when it reaches the tumor. Near-infrared-radiation triggers the collapse of hollow Au NPs to achieve Dox release. Moreover, two different breast cancer cell experiments showed that the D-P-HGNPs/21 sequential release system has a more effective inhibitory effect on tumor growth [28]. As a carrier of doxorubicin (Dox), Au NPs coupled with prostate-specific membrane antigen (PSMA) aptamer could specifically bind to target prostate cancer cells that overexpressing PSMA antigen. Dongkyu Kim et al. combined Au NPs, dox, and PSMA aptamer together, and then they observed that the assembled Au NPs complex were significantly more potent against targeted LNCaP cells that overexpressed PSMA antigen than against nontargeted PC3 cells that did not express detectable PSMA [29]. Daiki et al. conducted a similar study in 2021, confirming the targeting ability to LNCaP cells of this assembled Au NPs complex [30], and it was concluded that PSMA aptamer coupling Au NPs with adriamycin could kill LNCaP cancer cells more effectively than nontargeting PC3 cells [29]. In addition, Au-NPs-mediated gene therapy may significantly promote the improvement of anticancer therapy, and using Au NP compositions for effective RNAi delivery to silence protooncogene is a promising strategy for tumor treatment [31]. The experiments of Conde et al. in human cell models in vitro and vertebrate models have proved that this strategy can silence the c-myc protooncogene [32]. Although there are many advantages to gene therapy, drug delivery, and the photothermal triggered release based on gold nanoparticles, the results need further analysis and research to obtain practical results [31]. Gold nanoparticles coated with protein have also attracted people’s attention due to their excellent biocompatibility. Lysozyme-coated AuNPs synthesized by chemical reduction and collagen-coated AuNPs synthesized by the chemical reduction method were internalized efficiently by MG-63 osteosarcoma cells; thus, thus these two effectively absorbed coated nanoparticles may be used as diagnostic and therapeutic agents for osteosarcoma [33].

In addition, many studies have revealed that the photothermal effect of Au nanoparticles (Au NPs) and their composite nanoparticles is promising in the clinical application of tumor therapy [34]. Au NPs possess a surface plasmon resonance (SPR) effect, high biocompatibility, and stability in vivo; Au NPs are also ideal photothermal conversion materials for increasing optical absorption directly [35]. In vitro infrared ray and laser irradiation on the tumor area can directly kill tumor cells by causing local high heat through the accumulation of photothermal conversion agents (PTAs), such as Au NPs in the tumor [36]. This was called photothermal therapy (PTT); PTT induces less damage to normal tissues due to the fact that cancer cells are more intolerant to heat compared with normal cells [37]. However, there are some drawbacks that impair the clinical result of PTT, such as inadequate tumor accumulation, low photothermal conversion efficiency, and the poor stability of PTAs [38]. Polydopamine (PDA)-coated Au-Ag NPs improve the photothermal efficiency by shifting the SPR peak to the 808 nm wavelength, which matches the most used laser in photothermal therapy [39], and PDA coating can improve biocompatibility, increase hydrophilicity, and reduce cytotoxicity [40]. However, the thermal effect of PDA-coated NPs and the excessive accumulation of PDA-coated NPs localized to lysosomes and mitochondria cause lysosomal dysfunction and oxidative stress, which can induce the autophagy of tumor cells and protect tumor cells from external stimulation to a certain extent [41,42]. Wang et al. found that tumor cells pretreated with the autophagy inhibitor chloroquine showed higher mortality, which also provide evidence to support this point of view [43]. Although autophagy has a tumor-protective effect to a certain extent, PDA-coated Au-Ag NPs still show a strong ability to kill tumor cells. In vitro cultured bladder cancer T24 cells showed that PDA-coated branched Au-Ag NPs could induce cell-cycle arrest and apoptosis of tumor cells through various mechanisms, and T24 cells treated with PDA-coated branched Au-Ag NPs could induce S phase arrest, which was associated with decreased cyclin A level and increased p21 level [44]. Cyclin A is protein involved in the initiation and termination of S-phase DNA replication in the nuclear [45]; p21 inhibits cell-cycle progression by inhibiting the activity of cyclin-dependent kinases [46]; and, consequently, T24 cells will undergo S-phase arrest. Moreover, Au-Ag NPs are also involved in intrinsic pathways’ induced apoptosis of tumor cells, and the BCL2 family controls the permeability of mitochondrial membrane [47]; PDA-coated branched Au-Ag NPs decreased the BCL2 level, which further led to the depolarization of mitochondrial membrane potential (ΔΨm) and the subsequent release of cytochrome c into the cytoplasmic matrix. This will activate caspase-8 and caspase-3 to trigger apoptosis in T24 cells [44]. In a xenograft mouse model treated by laser irradiation after the injection of high-dose NPs, tumor growth was significantly suppressed [44]. After 12 days of PTT, the cell morphology of the heart, spleen, and liver in mice did not change significantly, while the tumor cells showed significant nuclear lysis, suggesting the low toxicity of Au NPs to normal cells and specific high toxicity to cancer cells [44]. In addition, the mitochondrial accumulation of PDA-coated Au-Ag NPs in in vitro thyroid cancer cell models led to the inhibition of dihydroorotate dehydrogenase (DHODH) expression, followed by enhanced transcriptional activity of the p53 gene, and it induced the S-phase arrest of tumor cells [48]. The Figure 2 below shows the mechanisms of tumor proliferation cycle arrest and apoptosis induced by PDA-coated Au-Ag NPs under laser irradiation. Similarly, pH-responsive AuNPs (CytC/ssDNA-AuNP), by introducing a mixed layer of single-stranded DNA and cytochrome c, forms clustered particle clusters in acidic environments and thus has a low pH-specific high photothermal efficiency on near-infrared radiations [49]. These pH-responsive AuNPs can be employed for cancer targeted therapy and improve its photothermal conversion efficiency in PTT based on the acidic tumor microenvironment [50]. More importantly, the pH response behavior mechanism based on electrostatic interactions between particles makes it possible to reversibly aggregate or disassemble particle clusters according to the solution pH, suggesting that the CytC/ssDNA-AuNP has the potential to exert its therapeutic effect on lesions repeatedly [49].

### 3.2. Carbon Nanoparticles

Carbon has many nano-allotropic forms, including fullerenes, nanotubes, and nanodiamonds. Their rich forms bring them many possibilities in tumor recognition, treatment, and drug delivery [51]. However, the hydrophobicity of carbon nanoparticles limits their medical application. This problem can be solved to some extent by modifying hydrophilic groups on the surface of carbon nanoparticles, and the toxicity of carbon nanoparticles can be reduced at the same time [52,53]. Paclitaxel (PTX) is a commonly used chemotherapy drug for breast cancer, and the effect is affected by poor water solubility; thus, albumin-bound PTX is the common dosage form in the clinic to improve its water solubility [54]. Shao et al. [55] found that when human serum albumin (HSA)-modified single-walled carbon nanotubes (SWCNTs), namely SWCNT–HSA complex, were used as a carrier for delivering PTX, the cell uptake rate of MCF-7 breast cancer cells could reach 80%, and this complex showed a stronger antitumor effect than has-modified PTX. Diamond nanoparticles (NDs) have high biocompatibility and low toxicity to normal cells; they can also form complexes with poorly water-soluble chemotherapeutic agents [56]. The existence of the BBB is a huge obstacle in the treatment of malignant brain tumors [57]. Liang et al. suggested NDs labeled by PEGylated denatured bovine serum albumin (BSA), and tumor vasculature-targeting tripeptides RGD, which is dcBSA-PEG-NDs, showed high efficiency in selectively targeting tumor sites in U-87 MG-bearing mice through the BBB [58]; in vitro BBB models revealed that the transcytosis mechanism and an additional direct cell–cell transport via tunneling nanotubes are both involved in this process [59]. One way to silence defective genes is by delivering interfering RNA (siRNA) into tumor cells. However, the instability of siRNA and low uptake of tumor cells limit its application, but in vitro experiments of MCF-7 breast cancer cells suggested that modified NDs could be made into a complex with siRNA for more efficient delivery of siRNA to tumor cells and silencing the expression of defective genes [60].

Another application of carbon nanomaterials is to identify tumor tissues and normal tissues during surgery, as well as lymph nodes’ draining from tumor areas. Thyroid cancer is the most pervasive endocrine malignancy, with its incidence increasing dramatically in recent decades, and papillary thyroid cancer (PTC) is the major contributor, as it accounts for 85% to 90% of all thyroid cancers [61]. According to the data from 1994 to 2013, at least 28.6% of PTCs had tumors with a diameter of 1 cm or less, which is called papillary thyroid microcarcinomas (PTMCs) [62]. The standard surgical procedure for PTC is total thyroidectomy and appropriate lymph-node dissection; the main risks of this surgery are parathyroid damage and recurrent laryngeal nerve damage [63]. Carbon nanomaterials with a diameter of 150 nm enter lymphatic vessels and then are transported to regional lymph nodes, but they do not enter blood vessels [64]. The intertumoral injection of carbon nanomaterials to stain tumor tissues and regional lymph nodes to protect parathyroid tissues has been widely used [65]. However, recent research revealed that the intertumoral injection of carbon nanomaterials cannot decrease the incidence of hypoparathyroidism, and this may be associated with the fact that the parathyroid glands possess compensatory potential [64]. For low-risk endometrial cancer patients, clinical trials have confirmed that cervical injection of carbon nanoparticles combined with indocyanine green can significantly improve the detection rate of sentinel lymph nodes (SLNs) [66]. In patients undergoing laparoscopic radical gastric cancer surgery, those who received carbon nanomaterials labeling can increase the lymph node discovery rate by 25.7% and shorten the operation time by 15.3% compared with the control group [67]. PTMCs are considered low-risk cancers. As one of the most attractive techniques, thermal ablation (TA) has achieved great success in many malignant diseases, including liver cancer and kidney cancer. In recent years, this technology has been gradually introduced into the treatment of PTMCs and even PTCs [68], the effect of microwave ablation was satisfactory, and the tumor volume reduction rate (VRR) reached or even exceeded 99% [69]. Compared with conventional open surgical methods, the incidence of complications is relatively low, and the difference between the recurrence rates was not statistically significant [69]. However, microwave ablation still has a certain risk of damaging the surrounding tissue. Experiments in mice receiving thyroid cancer TPC-1 xenotransplantation showed that the intertumoral injection of carbon nanomaterials can absorb near-infrared light, converting light into heat and achieving a temperature of 50–56 °C in the tumor; this is sufficient to kill tumor cells, thus avoiding systemic toxicity and protecting the parathyroid gland because the carbon nanomaterials act only on tumor tissue [70]. This suggests that the application of the photothermal effect of carbon nanomaterials in thyroid cancer is a promising direction in the future.

### 3.3. Lipid-Based Nanoparticles

Lipid-based nanobiomaterials, the most widely studied anticancer drug-delivery systems, are vesicular structures consisting of a single lipid bilayer that encases the anticancer drug in its hydrophilic core [71]. Lipid-based nanobiomaterials mainly include liposomes, nanoemulsions, lipid nanoparticles (LNPs), and solid lipid nanoparticles (SLNs), which offer many advantages, such as easy fabrication, the ability to self-assemble in aqueous media, enhanced bioavailability, biocompatibility, biodegradability of major components, low toxicity, and the ability to carry hydrophilic and hydrophobic compounds [72]. In addition, the main problems that this type of drug-delivery system faces are how to increase their stability, how to increase the residence time in circulation, and how to improve the efficiency of delivery of the drug or nucleic acid molecules to the target cells [73]. PEGylated liposomes, or long-circulating (stealth) liposomes, are liposomes modified and functionalized with hydrophilic polymer chains, including polyethylene glycol (PEG), polyethylene oxide (PEO), poloxamer, poloxamine, polysorbate (Tween-80), and lauryl ethers (Brij-35); this can effectively increase their residence time in circulation [74]. Lipid-based drug-delivery systems function in two ways: One way is active targeting, as described previously, by targeting specific sites, coupling liposomes to ligands that could bind to specific target cell receptors [75]. The other way is passive targeting, which is realized through EPR effect [73]. Different schemes can be designed to alter the average nanometer size, homogeneity, surface potential, drug loading, and ligand type of lipid-based nanoparticles to enhance their drug-delivery efficiency to the targeted cells [76,77]. Research in recent decades has put various liposome formulations into biomedical applications or clinical trials after overcoming the abovementioned shortcomings, as is shown in Table 1 below.

#### 3.3.1. Advances in Multifunctional Lipid-Based Nanobiomaterials

Liposomes have been widely used to combine different therapeutic classes of anticancer drugs for chemotherapy and immunotherapy, gene therapy, photochemotherapy, etc. Liposomal doxorubicin (Doxil) is the first therapeutic nanoparticle to receive clinical approval for cancer therapy using PEGylated liposome, laying the foundation for intensive research in nanotechnology for tumor-targeted drugs [78]. Moreover, subsequently, many liposomal formulations for anticancer drug delivery have successfully entered clinical trials.

#### 3.3.2. New Therapeutic Techniques Involving Lipid-Based Nanobiomaterials

Recently, attention has been drawn to the study of natural liposomes. Exosomes, as membrane vesicles with a diameter of 30–100 nm, have a double lipid membrane with the same origin pathway as the plasma membrane, containing proteins and genetic material that play an important role in intercellular communication inside [79]. Due to their inherent excellent properties, including their wide distribution in biological fluids, inherent homing ability, and the ability to penetrate the BBB, exosomes can undoubtedly be ideal drug-delivery carriers [80]. For example, exosomes isolated from MSCs can be used as drug nanocarriers to load the paclitaxel (PTX) for pancreatic cancer treatment [81]; exosomes isolated from mouse immature dendritic cells (imDC) can be used to load the doxorubicin (Dox) for breast cancer treatment [82].

In addition to this, attempts have been made to unite multiple nanocarriers so as to overcome their respective drawbacks without compromising their original properties. Lipid–polymer hybrid nanoparticle (LPHNP) systems can be a robust drug-delivery podium with high encapsulation efficiency; defined release kinetics; excellent tolerable serum stability; and well-triggered tissue-, cellular-, and molecular-targeting properties [83]. In 2010, Wang et al. proposed an LPHNP for prostate cancer models, named ChemoRad NP, intended for the codelivery of chemotherapeutics and therapeutic radioisotopes; this platform is mainly composed of two parts, the polylactic-co-glycolic acid (PLGA) polymeric core, which encapsulates chemotherapeutic agent (docetaxel), and the DMPE-DTPA lipid chelator layer, which chelates radiotherapeutic agent (indium-111 or yttrium-90); they demonstrated the better delivery ability and higher therapeutic efficacy of ChemoRad NPs [84]. In 2013, Zheng et al. successfully synthesized PLGA–lecithin–PEG hybrid NPs, which inhibit DOX-sensitive MCF-7 cells and DOX-resistant MCF-7/ADR cells growth through doxorubicin (DOX) and indocyanine green (ICG) loaded in PLGA–lecithin–PEG nanoparticles (DINPs); these DOX/ICG-loaded lipid–polymer nanoparticles show faster DOX release, longer retention time in tumors, and improved chemo-photothermal behavior under laser irradiation [85].

### 3.4. Polymeric Nanoparticles

Polymeric nanoparticles (PMs) are particles in the size range of 10–1000 nm, formed by amphiphilic block copolymers that consist of hydrophilic and hydrophobic polymeric chains connected via covalent bonds [86]. PMs are spontaneously formed by the self-assembly of block copolymers when placed in an aqueous environment [86]. Moreover, anticancer drugs can be loaded into a polymer core or adsorbed on the surface of a polymer shell [87]. Nanocapsules are capsule systems in which the drug is confined within a cavity surrounded by a unique polymeric membrane, while nanospheres are matrix systems in which the drug is physically and uniformly dispersed in the matrix [86,88]. Biocompatibility, biodegradability, and nontoxicity are the main features of polymeric NPs. According to report from Maurya et al., the use of polymeric NPs is safe for humans [86]. The advantages of polymeric NPs as drug carriers include the fact that they can control the drug release rate, protect drugs and other biologically active molecules from environmental influences, and improve bioavailability and therapeutic index of drug [87].

When using polymeric nanoparticles as carriers of anticancer drugs, they are not usually destroyed by phagocytes in circulation, but sequestration occurs in MPS-enriched organs. If the polymeric nanoparticles are not biodegradable, the particles will accumulate in these organs, most commonly in the liver and spleen, eventually leading to toxicity and other negative side effects [89].

Hadia et al. developed a polymeric nanoparticle of chitosan-encapsulating docetaxel- cyclodextrins successfully with ionic gelation method; they figured that this DTX CDs/CS polymer nanoparticle shows considerable advantages in drug release compared to pure the drug [90]. In addition, they assessed the safety of the PM by giving oral CDs to rabbits. They found that sulfobutylether β-cyclodextrin (BE7-β-CD) and other kinds of CDs remain intact and are nearly nonabsorbent in the gastrointestinal tract; CDs that enter the circulation are excreted only by the kidney [90]. They showed minimal reversible toxicity to the kidneys, liver, and lungs which depends on the dose and duration of administration [90]. Another recently studied PM is L-glutamic acid-g-p (HEMA) polymeric nanoparticle; a study in 2020 used the human bronchial epithelial cell line (16 HBE) and human monocytic cell line (THP-1) cultured in vitro to perform the cell migration test for a wound-healing study. They observed 21% percentage closure difference between control and exposure groups at 2 h; this indicates that the cell proliferation and migration of exposed cells were slower than control cells [91]. Although some studies suggested that amines in poly L-glutamic acid in nanoparticles can affect toxicity, which may induce hemolysis [92,93], they found no effect of HEMA on red blood cells from rabbit fresh blood samples [91]. The hen’s egg test/chorioallantoic membrane test is the standard test in vitro study of ocular irritation for alternatives to animal testing [94], with intravenous administration of HEMA, and no hemorrhage, vascular lysis, or coagulation effects were shown, thus hinting that HEMA-based nanoparticles are a safe ocular drug-delivery system [91]. However, due to the limitations of their research methods, their experiments were not able to assess the toxicity of HEMA in vivo truly and roundly. Poly-alkyl cyanoacrylate (PACA) has good biodegradability and a high loading capacity, making it a promising drug carrier [95]. Einar et al. noted that PACA has a certain cytotoxicity which was associated with the degradation rate. PBCA, PEBCA, and POCA are the three types of PACAs. PEBCA NPs with an intermediate degradation rate were significantly less toxic than both PBCA and POCA NPs (fast and slow degradation rate). This toxic effect may be related to the aggregation and perinuclear localization of intracellular lysosomes induced by PACA [96]. In Table 2, we summarized the studies on toxicity experiments of different polymer nanoparticles.However, more extensive studies are needed to clarify the true metabolic process and toxicity of polymers in vivo. Taken together, according to the existing evidence as shown in Table 2 which summarized the studies on toxicity experiments of different polymeric nanoparticles, polymer-based nanoparticles have good biocompatibility and low toxicity; thus, they have a broad prospect for targeted drug transport.

#### 3.4.1. Functionalization of Polymeric Nanoparticles

The polymerization modification of PMs by chemical methods is the most common method of functionalization. Some examples of shielding groups involved in chemical modification include polysaccharides, polyacrylamide, poly (vinyl alcohol), poly (N-vinyl-2-pyrrolidone), PEG, and copolymers containing PEG, such as polyoxamine, polyoxamine, polysorbitol, and PEG copolymers [97]. Of all polymers tested so far, PEG and copolymers containing PEG are the most effective and commonly used method [97]. These polymers are usually highly hydrophilic and charge neutral, and they can help shield even hydrophobic or charged particles of blood proteins [89].

Shi et al. constructed FA-PEG/PEO-PPO-PCL mixed micelles loaded with DTX to enhance the targeting specificity of DTX and improve the anticancer efficiency [98]. Palanikumar et al. designed pH-responsive hybrid ATRAM-BSA-PLGA NPs, which are composed of a cross-linked bovine serum albumin shell and encapsulated PLGA core, and the shell is functionalized by acidity triggered rational membrane (ATRAM) peptide [99]. The mixed polymeric micelles have an ideal size and low CMC value and negligible hemolytic activity, which, respectively, ensure their good accumulation in tumor tissue, high circulation stability, and good biocompatibility [100,101].

To date, PEGylation is still the benchmark for the development of functional nanocarriers in drug-delivery systems. Nevertheless, it is interesting to note that many reports indicate that PEG nanoparticles can cause unexpected immunogenic reactions, which lead to “accelerated blood clearance (ABC) phenomenon” and hypersensitivity reactions (HSRs) [102]. The specific mechanism is shown in Figure 3 below. After repeated injection of PEGylated liposomes in rats, the hepatic accumulation increase of second-dose injection hints at the important role hepatic plays in the accelerated clearance [103]. In the meanwhile, this accelerated clearing phenomenon is also observed in normal rats who receive transfusion of rat’s serum treated with PEGylated liposomes injection [104], so cellular immunity (Kupffer cells) and humoral immunity work together, resulting in this ABC phenomenon [103,104]. This drawback of PEGylation has attracted a lot of attention because it brings potential challenges to clinical work, reducing the therapeutic effect of encapsulated drugs after repeated administration [102]. After the first injection of PEGylated liposomes, they bind to B cells in the splenic marginal zone, triggering the production of anti-PEG IgM antibody in a manner independent of T cells [105]; this will enhance the uptake of PEGylated liposomes by hepatic Kupffer cells in the second dose of PEGylated liposomes injection, leading to an increase in the clearance rate [106,107]. The carrier structure may affect the ABC phenomenon. Xu et al. modified liposomes with cleavable PEG-lipid derivatives (PEG-CHEMS and PEG-CHMC), and only a slight ABC phenomenon was induced [108]. Similarly, gadolinium-containing PEG-poly(L-lysine)-based polymeric micelle induced no ABC phenomenon, while PEGylated liposome induced a strong ABC phenomenon [109]. Maitani et al. pointed out that the micelle hydrophobic core or lipid bilayer of PEGylated liposome plays an important role on this phenomenon [109]. The drug encapsulated in the carrier seems to be an important factor that may affect the ABC phenomenon. In clinical application, repeated injection of PEGylated liposomal doxorubicin reduces its clearance; one study attributed this to the toxic activity of doxorubicin on the hepatic and splenic reticuloendothelial system [110]. Ishida et al. proved that encapsulated doxorubicin reduces the production of anti-PEG IgM by interfering with the proliferation of B cells and subsequently reducing immune response against PEGylated liposomes [111]. This could be verified in repeated injections of PEGylated liposomal topotecan in rats which induce a strong ABC phenomenon [112], because as a cell-cycle phase-specific drug, topotecan can only inhibit a fraction of B lympholeukocytes in the S phase of the cell cycle [112]. The injection of PEGylated nanocarriers can interact with the immune system and result in undesirable HSRs, also known as C-activation-related pseudoallergy (CARPA) [113]. Although the exact mechanism of PEG-induced HSRs has not been fully elucidated, there is growing evidence showing that complement activation plays a role in the process [114,115]. The complement system releases C3a and C5a, which lead to the activation of inflammatory cells, such as macrophages, basophils, and mast cells, and promote the occurrence of HSRs [114,115,116]. Recent research highlights the important role of anti-PEG antibodies in the PEG-induced CARPA classic pathway at least for the case of Pegfilgrastim (PEG-G-CSF) and PEGylated liposomes [117,118]. The use of other alternative chemical groups such as PDX [119], poly(amino acid)-based biodegradable polymers [120], and polycaprolactone containing sulfobetaine [121] have been proposed, but a better strategy is to minimize the immunogenicity of PEG; the use of increasingly branched PEGs (i.e., hyperbranched, star, dendritic, and bottlebrush) of lower per-branch molecular weight may diminish recognition by backbone-specific antibodies, while still maintaining the advantages of PEG [122]. Compared to the methoxy (OCH3), which is a common terminal group of polymers used in clinical applications, hydroxyl PEG-modified liposomes (PL-OH) efficiently reduced the anti-PEG IgM response in vitro [123]. Other attempts to modify terminal polymer groups include zwitterionic, ethoxy, and n-butyl ether [124,125]. Khanna et al. reported that pretreatment with mycophenolate mofetil, a B/T cell immunosuppressant, significantly improved treatment outcomes in a Phase I trial of patients with gout receiving pegloticase; this suggests that pretreatment or conjugation with an immunosuppressant may diminish the polymer immunogenicity [126]. Further research is urgently needed to improve our poor mechanistic understanding of polymer-induced immunogenicity and its associated short- and long-term health risks.

Compared with other nanomaterials, polylactide (PLA) and polylactic-co-glycolic acid (PLGA) are the most promising polymeric candidates for drug-delivery systems, as they have low toxicity and are biodegradable, exhibiting good biocompatibility [127]. By controlling the size, shape, molecular weight, and the L:G ratio, PLA and PLGA can obtain ideal pharmacokinetic characteristics [128,129]. PLA can be also assembled as a hydrophobic block with other polymers, such as PEG, to produce amphiphilic block copolymers [100]. In the drug-delivery system, the stealth property is endowed by hydrophilic corona of amphiphilic copolymer micelles, and this reduces their uptake by reticuloendothelial system; this will prolong the lifetime of a loaded drug in the blood, improving the bioavailability [130]. In one study, the graphene oxide/PLA–PEG composites constructed showed a satisfactory paclitaxel loading capacity and drug-release performance, and this complex could enter the A549 cancer lung cancer vitro cells model and exhibited good cytotoxicity [131]. PLA-based nanoparticles show special advantages on alternative routes of administration (e.g., oral, pulmonary, and mucosal), as well as prolonged gene delivery efficacy [132,133]. In an MDA-MB-435s (cancer cells) murine xenograft model, the small interfering polo-like kinase 1 (siPlk1) delivered by PEG–PLA nanoparticles showed good results, which suppressed the tumor growth significantly [134]. The application of a B6-NP-encapsulated neuroprotective peptide, NAPVSIPQ (NAP), in the Alzheimer’s disease mouse model can produce excellent amelioration even at low doses [135]. This study shows that B6-peptide-modified NP can overcome the blood–brain barrier and has potential application in the targeted drug-delivery system for brain tumors [135]. In addition, lv. et al. reported a novel nanocomposite of polylactide (PLA) nanofibers and tetraheptylammonium-capped Fe3O4 magnetic nanoparticles; they verified this nanocomposite could effectively facilitate the interaction of daunorubicin with leukemia cells and remarkably enhance the permeation and drug uptake of anticancer agents in the cancer cells [136]. By increasingly functionalizing the PLA-based amphiphilic copolymer micelles and nanoparticles, key requirements for drug delivery, including stealthiness, controlled drug release, and targeting properties, are being met. However, new PLA-based nanoparticles that have been modified or connected with other polymers, and other nanoparticle types are emerging endlessly; their effectiveness and security in clinical are still waiting to be verified.

#### 3.4.2. Clinical Application of Polymeric Nanoparticles

PMs and their functionalization have been extensively explored for the treatment of various types of tumors because of their interaction with the well-known and well-studied tumor microenvironment. Amongst the various micellar formulations, PTX PMs have proven to have good clinical efficacy in treating advanced stages of lung cancer [137], breast cancer [138], central nervous system cancers [139], and oophoroma [140]. Zhou et al. used the solid-phase synthesis method to construct a paclitaxel long circulating nanoliposome targeting lung cancer and carried out experiments with tumor-bearing nude mice [137]. They found that this could improve the safety of paclitaxel administration and antitumor efficacy by improving the tissue distribution of paclitaxel in the body [137]. Shi et al. introduced ethoxy polyethylene glycol folic acid (FA-PEG) into the DTX-loaded micelles as an effective targeting part of the FA-PEG/PEO-PPO-PCL micelles prepared [98]. The experiments showed that it has excellent self-assembly ability in water and strong antidilution stability in circulation [98]. In addition, in vitro cytotoxicity results showed that FA-PEG/PEO-PPO-PCL micelles had higher cytotoxicity on FR-positive MCF-7 cells than PEO-PPO-PCL micelles [98]. Interestingly, Chen et al. reported a kind of cancer cell membrane hidden nanoparticle for the targeted delivery of doxorubicin and PD-L1 siRNA [141]. Through cell membrane stealth, biomimetic nanoparticles can be modified and functionalized through self-recognition and source-targeting capabilities, accompanied by long blood circulation and escape from immune capture, to achieve precise cancer targeted therapy [141].

### 3.5. Protein Nanoparticles

Proteins are naturally existing biomolecules that are considered to be ideal materials for nanoparticle preparation, owing to their safety, biocompatibility, and biodegradability [142]. This kind of nanoparticles from natural proteins can be metabolized by the body and easily surface modified with anticancer drug attachment and ligand binding [143]. Proteins can be divided into two types: animal proteins, such as albumin, gelatin, elastin, milk protein, and whey protein; and plant proteins, such as gliadin, soybean protein, and corn protein [144].

#### 3.5.1. Animal Proteins

Although the term “albumin” is often associated with serum albumin, it is also often used to describe a family of proteins characterized by being soluble in water [145]. In this family, serum albumin and whey protein are the most used proteins for preparing drug-delivery nanoparticles [146]. Two main members of the serum albumin group are human (HSA) and bovine serum albumin (BSA). HSA is widely used as a safe and effective carrier protein in different delivery systems due to the fact that it is a more nonimmunogenic plasma protein [146]. In Figure 4 below, we show different types of albumin-based carriers.

Albumin has two main binding sites, allowing for the complexation of metals, fatty acids and drugs including cisplatin, protein and peptide drugs [147]. The presence of these sites makes it possible to functionalize the specific delivery of a therapeutic moiety [148]. Albumin-based nanoparticles can be modified or loaded with targeting ligands on surface to change body distribution and improve their cellular uptake; this could avoid undesirable drug toxicity to some extent and improve the drug-targeting ability [148,149]. Wu et al. developed albumin copolymer micelles for delivery of doxorubicin; during the preparation, the hydrophobic interaction between polypeptide scaffolds is used to load doxorubicin; and compared with free DOX, this carrier shows higher drug cytotoxicity and higher PH dependent stability [150]. In addition to drugs, proteins can be also delivered by the BSA-based micelles to their target site. Jiang et al. prepared BSA-based polyionic complex micelles for Spry1 delivery. These micelles showed improved cytotoxicity on MCF7, which belongs to breast cancer cell lines, and exhibited high anticancer efficacy by inhibiting the growth of three dimensional MCF-7 multicellular tumor spheroids [151]. Moreover, in the past few years, various albumin-based nanoconjugates, including albumin–polymer conjugate, albumin–drug conjugate, and albumin–metal conjugate, have also been investigated. Cysteine (Cys) and lysine (Lys) residues of albumin are the most explored binding sites for preparing covalently conjugated albumin nanoconjugates; meanwhile, the noncovalent interaction between albumin and conjugation moiety is because of hydrophobic and electrostatic interactions, which also act as a driving force in the formation of albumin conjugates [152]. Because albumin has a long half-life in the human body, an albumin–drug conjugate prolonged the drug circulation in vivo, which also can ameliorate and overcome the multidrug resistance of anticancer drugs [153]. Several drugs, including DOX, cisplatin, docetaxel, etc., were used to prepare albumin–drug conjugates. Docetaxel–albumin conjugates were developed and verified by Esmaeili et al.; they possessed enhanced solubility and tumor targeting ability [154]. Similarly, the conjugate of SN38-HSA exhibited better solubility and stability [153]. Although the albumin-based nanodrug-delivery system has great advantages, the inherent disadvantages of albumin are its limited application, including intrinsic target groups lacking protein hydrolysis stability; and its limited and low applicability to hydrophilic and electrophilic drugs [155,156]. However, by conjugating albumin with another polymer, these drawbacks can be overcome. Compared to the polymer–platinum conjugate without albumin coating, the albumin-coated polymer–platinum conjugate can significantly increase the uptake rate of ovarian cancer cells and enhance their cell toxicity [157]. Moreover, as previously mentioned, Liang et al. suggested NDs labeled by PEGylated denatured bovine serum albumin (BSA) and tumor vasculature targeting tripeptides RGD, which is dcBSA-PEG-NDs, showed high efficiency in selectively targeting tumor sites in U-87 MG bearing mice through BBB [58]. In another study, Liu et al. prepared DOX-encapsulated cetuximab-functionalized BSA–PCL nanovesicle as a tumor-targeted nanocarrier, and they observed enhanced antitumor activity [158]. Tang et al. developed vitamin E (VE)–albumin core–shell nanoparticles for paclitaxel (PTX) delivery to improve the chemotherapy effect in MDR breast cancer models. Compared with NPs without VE (PTX NPs), PTX VE NPs significantly increased the cell uptake of PTX, showing stronger cytotoxicity and higher anticancer efficacy [159]. Shen et al. prepared the delivery system of hyaluronic acid and human serum albumin modified erlotinib nanoparticles (ERT-HSA-HA NPs) by using a precipitation method. An in vivo study found that the particles showed excellent anti proliferation effect on A549 cells. In terms of antitumor activity in vivo, ERT-HSA-HA-NP-treated mice showed significantly inhibited tumor growth and no recurrence after 30 days of treatment [160]. In conclusion, albumin-based nanocarriers have the advantages of being relatively safe and easy to prepare, the capability to deliver different types of molecules, and site-specific targeting by surface modification. These properties make albumin the most widely used protein for drug-delivery systems’ preparation.

#### 3.5.2. Plant Proteins

Compared with animal protein, plant protein has better biocompatibility and less immunogenicity [161]. Lee et al. developed chondroitin sulfate hybrid zein nanoparticles for the targeted delivery of docetaxel. Compared with free docetaxel, these NPs have improved pharmacokinetic properties and significantly enhanced antitumor efficacy, and the systemic toxicity caused can be ignored [162]. Gulfam et al. used an electrospray deposition system to synthesize gliadin and gliadin gelatin composite nanoparticles to control the delivery of anticancer drugs (such as cyclophosphamide) and regulate their release rate, and they proved that cyclophosphamide-loaded 7% gliadin nanoparticles can make breast cancer cells engage in apoptosis 24 h later [163]. More and more researchers are paying attention to protein nanoparticles; there are also many articles reporting the research progress of various protein nanoparticles. However, researchers still need to consider how to make more protein nanoparticles into clinical applications, not just research. In addition, the safety and effectiveness of protein nanoparticles as anticancer drug carriers need to be further determined by in vivo research.

## 4. Conclusions

In this review, we highlighted the recent advances in the development and application of nanoparticles in cancer treatment and summarized the advantages and disadvantages of the main nanoparticles in Table 3. Nanoparticles can be passively targeted toward tumor tissues due to their special size. However, with the continuous development of nano-engineering, various surface-modified nanomolecules can be more efficient and specifically targeted toward tumor cells. For example, many nanoparticles modified by specific ligands and monoclonal antibodies have been used in clinical trials and clinical drug targeting transportation systems. The accumulation of gold nanoparticles in tumor cells will induce tumor-cell-proliferation stagnation and apoptosis, but at the same time, it will also induce autophagy to improve the anti-external damage ability of tumor cells. Embedding anti-autophagic drugs in gold nanoparticles may be a solution. The photothermal effects of gold and carbon nanoparticles provide new possibilities for antitumor therapy based on surgery, radiotherapy, and chemotherapy, but their safety and availability need to be further studied and improved. Polymer nanoparticles and protein-based nanoparticles show outstanding biocompatibility and low toxicity. Liposome nanoparticles have been widely used in the clinic. Current studies and clinical trials focus on the modification of liposomes and the embedding of polymers to improve their effects. Liposomes loaded with new chemotherapy drugs and special RNA show different possibilities in the treatment of different tumors. The increasing surface-modified particles have improved their antitumor specificity to some extent in in vitro experiments; this also increases its instability and the risk of biological toxicity after metabolism in vivo. With the emerging and rapid development in nanomedicine, it will still be a huge challenge for us to balance biocompatibility and toxicity in the future.

## Figures and Tables

**Figure 1 pharmaceutics-15-00024-f001:**
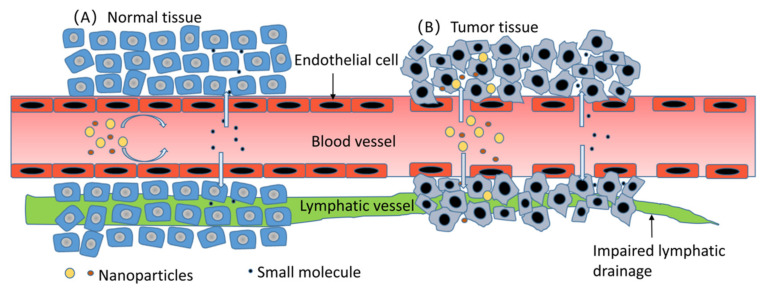
Compared with the tumor tissue in (**B**), the nutrient vascular epithelial cells in the normal tissue in (**A**) are arranged very closely, and only small molecules are easy to pass through; however, in (**B**), nanoparticles with a molecular diameter of 100–800 nm can smoothly pass through the nutrient vessels into the tumor tissue [4]. Moreover, there are few lymphoid tissues or only drainage-damaged lymphatic vessels in the tumor tissue, and the nanoparticles’ clearance will also be reduced [5].

**Figure 2 pharmaceutics-15-00024-f002:**
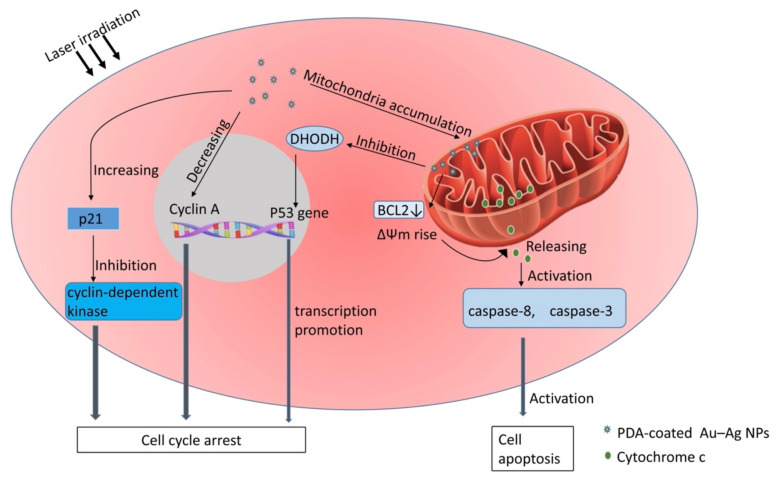
Mechanism of laser-irradiation-induced proliferation arrest and apoptosis of PDA-coated-Au–Ag-NP-treated tumor cells. In tumor cells, PDA-coated Au-Ag NPs increase the p21 level, which inhibits the activity of cyclin-dependent kinases, and decrease the Cyclin A level, which plays an important role in the initiation and termination of S-phase DNA replication at the nuclear. In addition, the mitochondrial accumulation of PDA-coated Au-Ag NPs leads to the inhibition of DHODH expression, followed by enhanced transcriptional activity of p53 gene, an important tumor-suppressor gene. All of these together will result in the S-phase arrest of tumor cells. The BCL2 family controls the permeability of mitochondrial membrane. PDA-coated branched Au–Ag NPs induced the decrease of BCL2 level, which further led to the depolarization of mitochondrial membrane potential (ΔΨm) and the subsequent release of cytochrome c into the cytoplasmic matrix; this activates caspase-8 and caspase-3, which then trigger cell apoptosis.

**Figure 3 pharmaceutics-15-00024-f003:**
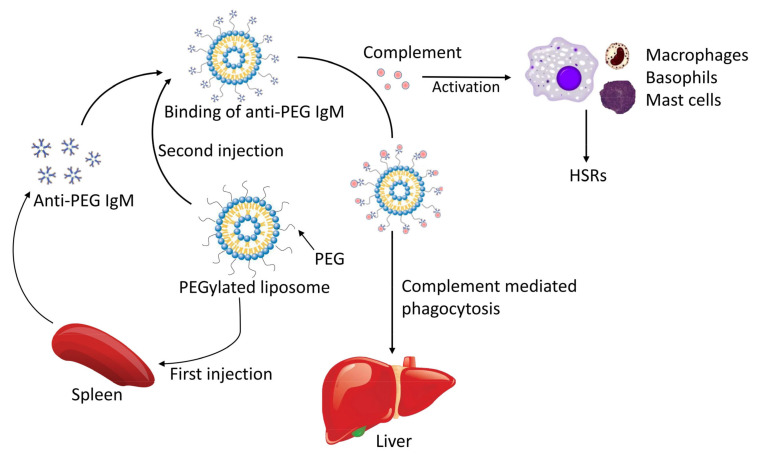
The mechanism of ABC phenomenon and HSRs induced by PEGylated liposomes for its immunogenicity.

**Figure 4 pharmaceutics-15-00024-f004:**
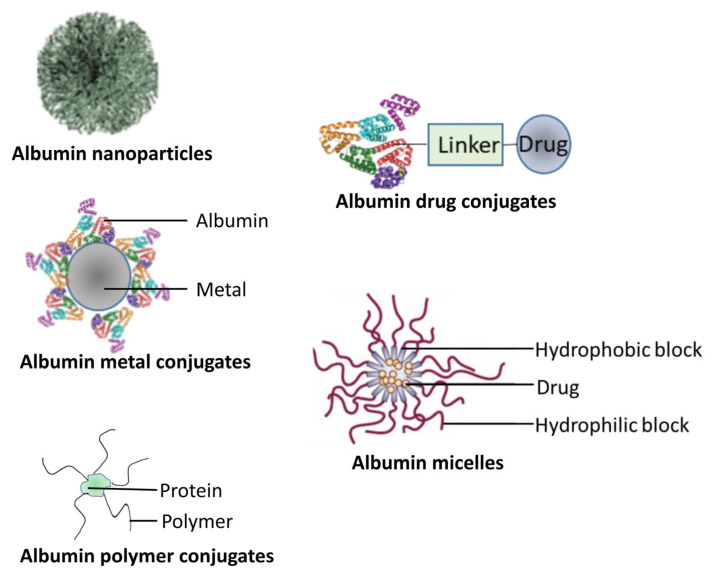
Different types of albumin-based carriers.

**Table 1 pharmaceutics-15-00024-t001:** Current lipid nanoparticles for cancer treatment and drug regimens being studied in clinical trials. Date from ClinicalTrails.gov.

Nanoparticle Formulation and Combination Drug Regimen	Type of Cancer	Clinical Trial ID	Phase
Liposomal doxorubicin combination with anti-CD47, ALX148, and pembrolizumab	Recurrent Platinum-Resistant Ovarian Cancer	NCT05467670	2
Lyso-thermosensitive liposomal doxorubicin (LTLD)	Relapsed/refractory solid tumors in children	NCT02536183	1
Liposome-encapsulated daunorubicin–cytarabine	Refractory Acute Myeloid Leukemia	NCT04049539	2
PEGylated liposomal doxorubicin and SL-172154	Ovarian cancers	NCT05483933	1
Lysosomal-associated membrane protein (LAMP) mRNA-loaded DOTAP liposome vaccine	Pediatric High-Grade Gliomas (pHGG) and Adult Glioblastoma (GBM)	NCT04573140	1
Liposomal HPV-16 E6/E7 multipeptide vaccine PDS0101	Oropharyngeal carcinoma	NCT05232851	1
PEGylated liposomal doxorubicin hydrochloride	Female reproductive system tumors	NCT04092270	1
Nanoliposomal irinotecan combine with trifluridine and tipiracil hydrochloride	Advanced gastrointestinal cancer	NCT03368963	2
PEGylated liposomal doxorubicin (PLD)	Advanced solid malignancies	NCT04244552	1
Liposomal Bcl-2 antisense oligodeoxynucleotide	Relapsing acute myeloid leukemia	NCT05190471	1
Vincristine sulfate liposome	Acute lymphoblastic leukemia	NCT02879643	1
Liposome-encapsulated daunorubicin–cytarabine	Advanced myeloproliferative neoplasms	NCT03878199	2
Irinotecan liposome and bevacizumab	Platinum-resistant fallopian tube, ovarian, primary peritoneal carcinoma	NCT04753216	2
Liposome-encapsulated daunorubicin–cytarabine combine with gemtuzumab ozogamicin	Relapsed or refractory acute myeloid leukemia (AML)	NCT03672539	2
BP1001-A (liposomal Grb2 antisense oligonucleotide)	Advanced or recurrent solid tumors	NCT04196257	1
CDX-301 and CDX-1140 in combination with PEGylated liposomal doxorubicin (PLD, Doxil)	Metastatic triple negative breast cancer	NCT05029999	1
liposomal irinotecan (nanoliposomal irinotecan (Nal-IRI))	Neuroendocrine neoplasms of the digestive tract	NCT03736720	2

**Table 2 pharmaceutics-15-00024-t002:** Experimental studies on toxicity of polymeric nanoparticles.

Biological System	Administration Route	Type of PMs	Mechanism of Toxicity	The Main Factors Affecting Toxicity	Reference
Rabbits	Incubation	CDs	Reversed kidney vacuolization; not toxic to heart, liver, spleen, and lungs	Dose and duration of administration	[90]
16 HBE and THP-1 cultured in vitro,	Straight way	HEMA	Slowing wound healing by affecting cell proliferation, migration, and uptake or ocular vascular occlusion	size and morphology of NPs	[91]
Fresh blood samples from rabbits,	Straight way	HEMA	No promotion of hemolysis		[91]
egg chorioallantoic membrane	intravenous	HEMA	No promotion of ocular vascular occlusion		[91]
Cell model cultured in vitro	Straight way	PACA	Lysosomal clustering and perinuclear localization	Degradation rate	[96]

**Table 3 pharmaceutics-15-00024-t003:** Major nanoparticles involved in this review and their features.

Types of Nanomaterials	Main Features	References
Au NPs	Stability, low toxicity. Gene therapy, protooncogenes silencing. Ideal photothermal conversion materials.	[31] [35]
Au-Ag NPs	Photothermal therapy. Inducing cell-cycle arrest and cell apoptosis.	[41] [44,48,49]
Carbon nanomaterials	Rich forms including fullerenes, nanotubes and nanodiamonds. Low toxicity, high biocompatibility. Specific recognition of tumor tissue during treatment and surgery. Photothermal therapy.	[56] [64,66,67] [70]
Lipid-based nanoparticles	Widely used, high biocompatibility, biodegradability	[72]
Polymeric nanoparticle	Safe, low toxicity.	[86]
PEG-based nanoparticles	Drug half lifetime prolonged. Immunogenicity inducing ABC phenomenon	[102,103,104]
PLA-based nanoparticles	Biodegradability, low toxicity.	[127]
Albumin-based nanoparticles	Drug half-lifetime prolonged. Development of various conjugates	[153] [152]

## Data Availability

The data presented in this study are available upon request from the corresponding author.

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
