# Peer review of "Novel Development of Nanoparticles—A Promising Direction for Precise Tumor Management"

_pharmaceutics, 2022, doi:10.3390/pharmaceutics15010024_

Round 1

Reviewer 1 Report

In this manuscript, the authors summarized several kinds of nanoparticles for passive and active tumor-targeting. Overall this article provides useful information of nanoparticles for cancer therapy. However there are some issues to be addressed before further consideration of the acceptance of publication.

1.     A summary scheme should be drawn to illustrate the nanomaterials mentioned in this review manuscript.

2.     In section 1.2.6, the authors mentioned MMP-2 for active tumor targeting. Although MMP-2 is highly expressed in many types of cancers, it is mostly secreted rather than staying on the cell membrane. Therefore it might not be an ideal target for active tumor targeting. The authors might consider removing it from this session.

3.     The titles of tables should be put on the top of the table rather than at the bottom.

4.     The authors described a lot about lipid nanoparticles, polymeric nanoparticles, and protein nanoparticles. It would be better if the authors can add figures that illustrate one or two important studies to each of these sessions. It will make the manuscript more readable.

5.     I noticed there were lots of DOIs in the References. The authors should revise them to be compatible with the formatting of this journal.

Author Response

Thank you for the comments, we have revised the manuscript according to your comments.

  1. We briefly described the major nanoparticles involved in this review and their features in Table3 before the conclusionpart.
  2. We have reviewed a large number of research articles on MMP-2, MMPs areoverexpressed in some cancers, which play critical role in the degradation of basement membranes and the extracellular matrix. Its protease activity can be used to release an anticancer agent from a macromolecular carrier.The incorporation of stimulus-responsive moieties (the MMP-2 substrates including cleavable peptides) into the targeted delivery systems may achieve precise targetability and controlled release of drug molecules resulting in the improvement of their therapeutic profiles, but as you mentioned, MMP-2 mainly exists in the tumor matrix rather than on the surface of the tumor cell membrane, and in the current research on nano drug delivery systems, MMP-2 mainly plays the role of "release" rather than "targeting", so we removed this paragraph.
  3. 3. We put the titles of tables at the top of the table as requested.
  4. 4. In the latest manuscript, we have drawn the Figure3 showing the basic mechanismof ABC phenomenon and Figure 4 to make the paper more readable.
  5. 5. We downloaded the new endnote style from the website, and the current format is compatible with the journal.

Reviewer 2 Report

This. is an excellent review of the very large area. Since it is not possible to cover the whole area, the authors should at lease refer to a 2019 review on the whole area. DOtl:10.1080/21691401.2019.1577885   Artificial Cell, Nanomedicine, Biotechnology 2019. 47:997 

In addition, they should add much more details on the following:

1. PEG is a widely used approach, The authors only mentioned very briefly regarding its immunologic problem. They should add much more details. Furthermore, they only mentioned very very briefly attempts to solve this proble. They should add much more details about these

2. Polylactide and its copolymers have been extensively use as drug delivery system, yet the authors barely mentioned these. They should add much more pros and cons

3. On the topic of protein nanoparticles. There has been extensive developments expecially in Nanobiotherapeutics and much more details should be included. Details can be found in a 2021 open access 1,042 multiauthor book that can be found on www.artcell.mcgill.ca 

Author Response

Thank you for your suggestions, which is of great help to our current work. We have a deeper understanding of the initiation and development of nanotechnology in medicine.

  1. 1. We have added a discussion on the immunogenicity of PEG and its potential solutionsas requested.
  2. 2. We have added a paragraph about polylactide and its copolymers, and introduced the advantages of PLA-based nanoparticles in drug delivery systems after modification and assembly.
  3. 3. Thanks for the website you recommended. We carefully reviewed the relevant literature, so we focused on the development of albumin based nano carriers in the part of protein based nanoparticles intherevised manuscript. 

Reviewer 3 Report

The title of article should be revised as you directly write the title of issue, it should be specific.

similar work is already published 

Niculescu, A. G., & Grumezescu, A. M. (2022). Novel Tumor-Targeting Nanoparticles for Cancer Treatment-A Review. International journal of molecular sciences23(9), 5253. https://doi.org/10.3390/ijms23095253

Author Response

Thank you very much for the comments, we have made appropriate changes to the title of the article. In addition, we have carefully read the literature you recommended. Our article is similar to it in some aspects, but we focus on describing the mechanism of nanoparticles for targeted cancer therapy, and present possible drug delivery targets in as much detail as possible. In addition, we also enumerate and demonstrate different types of nanomaterials in drug delivery and other therapeutic effects. We described in detail the mechanism of photothermal therapy based on Au nanoparticles that may induce tumor cell apoptosis and its great potential in the future, as well as the great changes that carbon nanomaterials may bring in cancer treatment, including thyroid cancer. Thank you for your comments, which made us realize the shortcomings of our work. We hope the revised manuscript will contribute to this field.

Reviewer 4 Report

The review by Chen et al. highlights the importance of gold nanoparticles for tumor therapy. Overall, the review read well and is publishable after minor changes.

1.       Line 213: Carbon nanoparticles have been stated in many places but not identifies. There are many types of nanomaterials, carbon dots, diamond and many more. Authors must be careful while using nanoparticles/nanomaterials

2.       Line 221: Carbon nanoparticles with diameter of 150 nm. Nanoparticles should have 100 nm or below. Should you call them nanomaterials not nanoparticles?

3.       Line 170: AuNPs not Au-NPs. Check the rest of the review.

4.       In the gold nanoparticles section 2.1. the discussion is so random and not focused on specific examples. Similarity in many of the nanoparticle types.

5.       Section 2.1. Gold nanoparticles: It is mainly focused on gold nanoparticles not Au-Ag nanoparticles.

6.       Please add the general/specific references to the review:

DOI: 10.1016/j.nantod.2018.12.006, doi.org/10.3390/cancers14215366, doi.org/10.1021/acs.langmuir.0c01443, DOI: 10.1038/nrc.2016.108, DOI: 10.1038/s41598-019-56754-8

Author Response

Thank you very much for the comments, we have followed your suggestions to revise the manuscript.

  1. There are many kinds of carbon nanomaterials. We introduced diamond nanoparticles with a diameter of 4-5 nm, carbon nanotubes with a diameter of 1 nm, C60 fullerenes with a diameter of 0.7 nm, and we called them by nanoparticles.
  2. Thank you very much for the correction, we have modified the terminology appropriately in section 2.2, replacing carbon nanoparticles with carbon nanomaterials.
  3. Thank you very much for the correction, we have modified the terminology in manuscript, replacing Au-NPs with Au NPs.
  4. 4.According to your suggestions, we have made slight adjustments to the examples listed above, and added some examples introducing different therapeutic effects of gold nanoparticles in section 2.1, such as gold nanoparticles used as cancer targeted drug carriers .
  5. 5.Thank you for your suggestions, pointing out thatthe focus of this section is primarily on gold nanoparticles rather than gold-silver nanoparticles. But we did not modify the Au-Ag nanoparticles, because we believe that the mechanism of this part is relatively novel and has a positive impact on the development prospect of Au based nanoparticles.
  6. 6.We have obtained new inspiration from several of these articles and quoted them in our article (citations in the latest manuscript of 3, 28, 29, 30 and 44) .

Round 2

Reviewer 3 Report

Now its ok

Author Response

Thank you very much for your approval. We are very glad to see that this manuscript has improved a lot compared with the original manuscript after modification according to your suggestion.